# Optimized Preparation of Levofloxacin Loaded Polymeric Nanoparticles

**DOI:** 10.3390/pharmaceutics11020057

**Published:** 2019-01-30

**Authors:** Manuel López-López, Angela Fernández-Delgado, María Luisa Moyá, Daniel Blanco-Arévalo, Cecilio Carrera, Rafael R. de la Haba, Antonio Ventosa, Eva Bernal, Pilar López-Cornejo

**Affiliations:** 1Department of Chemical Engineering, Physical Chemistry and Materials Science, Faculty of Experimental Sciences, University of Huelva, Campus de El Carmen, Avda. de las Fuerzas Armadas s/n, 21071 Huelva, Spain; manuel.lopez@diq.uhu.es; 2Department of Physical Chemistry, University of Seville, C/Profesor García González 1, 41012 Seville, Spain; angelafd94@gmail.com (A.F.-D.); moya@us.es (M.L.M.); danblaare@gmail.com (D.B.-A.); evabernal@us.es (E.B.); 3Department of Chemical Engineering, University of Seville, C/Profesor García González 1, 41012 Seville, Spain; cecilio@us.es; 4Department of Microbiology and Parasitology, University of Seville, C/Profesor García González 2, 41012 Seville, Spain; rrh@us.es (R.R.d.l.H.); ventosa@us.es (A.V.)

**Keywords:** levofloxacin, nanoparticles, PLGA, chitosan

## Abstract

In this work, poly(lactic-*co*-glycolic acid) (PLGA) and chitosan (CS) nanoparticles were synthesized with the purpose of encapsulating levofloxacin (LEV). A thorough study has been carried out in order to optimize the preparation of LEV-loaded polymeric nanoparticles (NPs) suitable for parenteral administration. Changes in the preparation method, in the organic solvent nature, in the pH of the aqueous phase, or in the temperature were investigated. To the authors´ knowledge, a systematic study in order to improve the LEV nanocarrier characteristics and the yield of drug encapsulation has not been carried out to date. The physicochemical characterization of the NPs, their encapsulation efficiency (EE), and the in vitro release of LEV revealed that the best formulation was the emulsion-solvent evaporation method using dichloromethane as organic solvent, which renders suitable LEV loaded PLGA NPs. The morphology of these NPs was investigated using TEM. Their antimicrobial activities against several microorganisms were determined in vitro measuring the minimum inhibitory concentration (MIC). The results show that the use of these loaded LEV PLGA nanoparticles has the advantage of the slow release of the antibiotic, which would permit an increase in the time period between administrations as well as to decrease the side effects of the drug.

## 1. Introduction

Infectious diseases are recognized as a public health challenge [1]. The appearance of antibiotics, more than 70 years ago, has totally changed the treatment of infectious diseases, contributing to significantly diminish the associated mortality [2]. However, the antibiotic drug therapies used today have several limitations, such as inadequate drug dosage, important side effects, and drug abuse. It is for the above reasons that antimicrobial resistance has increased, now becoming a global concern. Without effective antimicrobial drugs, medical procedures such as major surgery, cancer chemotherapy, or organ transplantation become highly risky.

The use of nanocarriers as antibiotic delivery systems could help to improve the current limitations associated to antibiotic therapy. Antibiotics could be poorly soluble in water, sensitive to temperature, and easily degraded by proteolytic enzymes. They will be protected (encapsulated) inside the nanocarriers. The delivery of the drug can be controlled, and a slow release would permit an increase in the time period between administrations [3]. Furthermore, it has been shown that antibiotics encapsulated in nanocarriers had enhance antimicrobial activity against resistant bacterial strains [4,5]. Therefore, the use of nanocarriers can be considered as an excellent approach to avoid, at least partially, the limitations associated with antibiotic therapy. Until now, several studies have used nanosized delivery vectors for antibiotics such as ciprofloxacin, ampicillin, amoxicillin, vancomycin, etc. [2,6,7,8,9,10,11,12,13,14,15,16]. Frequently, the nanocarriers of choice have been polymeric nanoparticles, given that they can be prepared with many different polymers providing flexibility for designing the delivery systems. Moreover, there are a plethora of approaches for the preparation of polymeric nanocarriers. The selection of the nanoencapsulation method should take into account the drug nature, personnel safety, the nanoparticle requirements, etc. Methods such as emulsion–solvent evaporation/extraction, nanoprecipitation, salting out, emulsion solvent diffusion, ionic gelation, and many others have been used [2,17,18,19].

Natural polymers, such as alginate or chitosan (CS), and synthetic polymers, such as polycaprolactone (PCL) and poly(lactic-*co*-glycolic acid) (PLGA) are extensively employed. The most widely used polymer in the preparation of nanocarriers is PLGA [17,20,21,22]. It was approved by the United States Food and Drug Administration (FDA) as it is biodegradable and biocompatible, with no side effects. Among the natural polymers, CS is frequently used in the design of nanomedicines [18,19,23,24,25,26]. Its cationic character and its solubility in water, as well as its biocompatibility and low toxicity, are attractive characteristics for using CS in the preparation of nanocarriers.

Levofloxacin (LEV) is a third-generation fluoroquinolone antibacterial agent active against most Gram-positive and Gram-negative microorganisms. It has been shown to be efficient against several respiratory tract, obstetric, genitourinary, and skin tissue infections. LEV has been loaded in several types of polymeric nanoparticles. Ameeduzzafar et al. and Gupta et al. developed a formulation for the loading of LEV in CS nanoparticles for ocular delivery [27,28] and Guan et al. also prepared LEV loaded CS nanoparticles by using ionotropic gelation [29]. Kumar et al. [30] and Cheow et al. [31] prepared loaded LEV PLGA nanocarriers. The latter also used poly(ε-caprolactone) as a biocompatible polymer to prepare the nanoparticles. Jalvandi et al. [32] investigated the slow release of LEV conjugated on silica nanoparticles from PCL nanofibers. Umesh et al. [33] synthesized polyhydroxyalkanoate (PHA) nanocarriers for loading LEV by using a triple emulsion method. Although the encapsulation of LEV in polymeric nanocarriers has been previously investigated, a systematic study in order to improve the nanocarrier characteristics and the yield of drug encapsulation has not been carried out to date. With this in mind, the aim of this study was to optimize the preparation of PLGA and CS polymeric nanoparticles (NPs) for encapsulating LEV by changing several parameters in the formulations. For the PLGA nanocarriers, double emulsion-solvent evaporation (DESE) and nanoprecipitation (NANOP) methods were used. Whereas for CS NPs, ionic gelation, ionic gelatin (IG), and polyelectrolyte complexation/ionic gelation (PC/IG) methods were utilized. Considering literature data, in the case of PLGA, and taking into account the hydrophilic character of the LEV molecules, DESE and NANOP methods seem adequate in order to prepare NPs suitable for parenteral administration with a substantial encapsulation efficiency (EE). On the other hand, CS was chosen due to its positive charge. Since the physiological charge of LEV is −1, it was expected that CS nanocarriers could encapsulate the antibiotic. Furthermore, CS NPs obtained by the IG and PC/IG methods usually have the required size for parenteral administration. In this work, changes in the preparation method, in the organic solvent nature, in the pH of the aqueous phase, or in the temperature of preparation were investigated. The zeta potential, size, and EE of the variety of prepared nanoparticles were measured as well as the in vitro drug release. In the case of the more promising NPs, their antimicrobial activities were determined in vitro measuring the minimum inhibitory concentration, minimum inhibitory concentration (MIC), against several Gram-positive and Gram-negative microorganisms.

## 2. Materials and Methods

### 2.1. Materials

PLGA (RG 502 H), CS of low molecular weight (LMW, 19,000–50,000 Da), PVA (average mol wt. 30,000–70,000) and κ-carrageenan (Sigma, 22048-F), together with the rest of the reagents used were purchased from Sigma (Darmstadt, Germany). Pluronic F-68 was from Gibco (Madrid, Spain).

A buffered media of PBS pH = 7.4 was used in the spectrophotometric, microscopic, and release experiments. PBS tablets were obtained from Sigma-Aldrich (Darmstadt, Germany). One tablet dissolved in 200 mL of deionized water yields 0.01 M phosphate buffer, KCl 0.0027 M and NaCl 0.137 M sodium chloride, pH 7.4, at 25 °C.

Six bacteria were used in the antimicrobial studies. Gram-negative bacteria included *Escherichia coli* CECT 101, *Klebsiella pneumoniae* CECT 143, and *Pseu-domonas fluorescens* CECT 378. Gram-positive bacteria included *Enterococcus faecalis* CECT 481, *Mycobacterium phlei* CECT 3009, and *Staphylococcus epidermidis* CECT 231. These reference strains were obtained from the CECT (Spanish Culture Collection of Type Cultures, University of Valencia, Valencia, Spain).

### 2.2. Methods of Preparation

#### 2.2.1. Preparation of PLGA Nanoparticles

##### Double Emulsion-Solvent Evaporation Method (DESE)

Method A (DESE, dichloromethane)—The PLGA nanoparticles loaded with or without LEV were prepared by using the DESE technique [34]. 30 mg of PLGA was dissolved in 1 mL of dichloromethane (DCM) by magnetic stirring. The resulting solution was added dropwise to 6 mL of aqueous solution of 2% (*w/v*) PVA, containing 13 mg of LEV. The mixture was homogenized by sonication for 90 s using an ultrasonic processor Vibra Cell Sonics, model VCX 750 (Taunton, MA, USA, maximum net power output: 750 W) at a frequency of 20 kHz and an amplitude of 20% (30 s), 30% (30 s), and 40% (30 s).

The resulting emulsion was pipetted dropwise onto the solution of the same polymer mixture (30 mg of PLGA in 1 mL of DCM) to carry out a new emulsion. After a new session of ultrasonication following the same procedure, the final emulsion was poured over 6 mL of 2% (*w/v*) PVA in water. Finally, it was ultrasonicated for 5.5 min with amplitude of 20% (60 s), 30% (60 s), and 40% (210 s). 

The organic phase was evaporated by magnetic stirring of the suspension overnight, at room temperature. The polymeric nanoparticles loaded with LEV, PLGA/LEV, were recovered by ultracentrifugation at 12,000 rpm for 40 min. Then they were washed twice with water to remove the free drug. PLGA/LEV nanoparticles were dried under vacuum by using Eppendorf Concentrator plus for 40 min.

Method B (DESE, ethyl acetate)—PLGA/LEV nanoparticles were synthesized by a water-in-oil-in-water (w/o/w) DESE procedure [35]. First, a solution of 3 mL of water and 25 mg of LEV was added to 10 mL of ethyl acetate and 250 mg of PLGA, with a stir of 20,000 rpm by using an IKA T25 ULTRA-TURRAX dispenser (IKA, Staufen, Germany). This first w/o emulsion was poured to a solution of 50 mL of water and 0.3% (*w/v*) PVA, under mechanical stirring for 15 min at 20,000 rpm. Following this, this w/o/w emulsion was diluted over 100 mL of 2% *w/w* PVA by magnetic stirring. The organic solvent was evaporated maintaining the magnetic stirring (300 rpm) for 12 h. Then, PLGA/LEV NPs were collected by ultracentrifugation 1 h at 11,000 rpm and they were washed and dried as in the previous method. 

##### Nanoprecipitation (NANOP)

A method similar to one previously reported in the literature was used [36]. In presence of 10 mg of LEV, 100 mg of PLGA was dissolved in 5 mL of acetone. This solution was added dropwise to 20 mL of an aqueous solution of 1.5% (*w/w*) PVA under stirring at 3000 rpm by using an IKA T25 ULTRA-TURRAX dispenser. The organic solvent was evaporated maintaining the magnetic stirring (300 rpm) overnight at room temperature. The PLGA/LEV NPs were separated by ultracentrifugation at 11,000 rpm for 1 h. Finally, they were washed and dried using the procedure described above.

The method used to obtain the influence of organic solvent on the characteristics of the NPs and the LEV encapsulation was carried out using acetone (AC), tetrahydrofuran (THF), acetonitrile (ACN), DCM/ACN/Methanol 0.16/8.2/1.64, and ACN/Ethanol 4.5/5.5. However, the NPs yield obtained from the NANOP method using all the organic solvents, with the exception of acetone, was very low. For this reason, the NANOP method was only carried out with this organic solvent.

The method using acetone as organic solvent was repeated changing the surfactant nature: 10 mL of Pluronic 68 0.1% (*w/w*) in water.

In order to investigate the influence of pH on the encapsulation of LEV in PLGA NPs, this method of NANOP using Pluronic 68 as surfactant was repeated in the presence of HCl 10^−4^ M and of buffer Acetic/Acetate pH = 4.

#### 2.2.2. Preparation of CS Nanoparticles

##### Ionic Gelation (IG)

CS nanoparticles loaded with LEV were prepared using a modified procedure of Fan et al. [37,38]. CS was dissolved in an aqueous solution of acetic acid 0.2 mg/mL to obtain a solution of CS 0.5 mg/mL. This CS solution was stirred overnight by magnetic stirring (300 rpm) and the pH of the resulting solution was adjusted at 4.7 to 4.8 using an aqueous solution of 20% (*w/w*) NaOH. Following this, the CS solution was filtered with a syringe filter with a pore size of 0.45 µm. Tripolyphosphate (TPP) was dissolved in distilled water to prepare a solution of TPP 0.5 mg/mL and also filtered with a syringe filter with a pore size of 0.2 µm. 25 mg of LEV was dissolved in 30 mL of TPP 0.5 mg/mL maintained at 2 to 4 °C. 100 mL of CS 0.5 mg/mL was preheated at 37 °C in a water bath for 10 min and the solution containing TPP and LEV (30 mL, *T* = 2 to 4 °C) was added quickly under magnetic stirring at 700 rpm. The reaction was performed for 10 min. The CS/TPP/LEV NPs obtained were collected and dried following a procedure similar to that described above.

In order to study the influence of the pH on the encapsulation of LEV, the procedure described above was repeated adjusting the pH to 5.6.

##### Polyelectrolyte Complexation/Ionic Gelation (PC/IG)

CS/carrageenan nanoparticles loaded with the drug, CS/CRG/TPP/LEV, were obtained using a modification of the method proposed by Rodrigues et al. based on a polyelectrolyte complexation of CS with CRG and an ionic gelation of CS with TPP anions [39]. CS was dissolved in an aqueous solution of 1% (*w/w*) acetic acid to obtain a solution of CS 1 mg/mL. The stock solutions of TPP and CRG were prepared in pure water with a concentration of 2.5 and 10 mg/mL, respectively. In regard to LEV, 4.8 mg were dissolved in a mixture of 1 mL of TPP 10 mg/mL and 4 mL of CRG 2.5 mg/mL. The solution containing the drug was added dropwise to 40 mL of CS 1 mg/mL at 250 rpm and the NPs were formed spontaneously. The volumes used of each stock solution were chosen to obtain a CS/CRG/TPP ratio 4/1/1. The CS/CRG/TPP/LEV NPs obtained were collected and dried following a procedure similar to that described above.

Each preparation method was carried out three or four times.

### 2.3. UV-Visible Spectroscopy Measurements

A Shimadzu UV-1800 spectrophotometer (Shimadzu, Kyoto, Japan), connected to a water flow Lauda cryostat was used. A standard quartz cell of 1 cm path length was utilized for the absorbance measurements at 298 ± 0.1 K. The stability of the LEV solutions at pH = 7.4 was investigated recording the antibiotic absorbance spectrum at predetermined intervals and measuring the absorbance at 288 nm, the wavelength of the maximum absorbance. The results indicated that LEV solutions were stable for more than one day. The authors checked that in the absence of light the LEV was stable for more than two weeks.

### 2.4. Size and Zeta Potential

The zeta potential, ξ, and size distribution of the nanoparticles were determined using a Zetasizer Nano ZS Malvern Instrument Ltd. (Worcestershire, UK). The samples were diluted with filtered water to an adequate concentration. A scattering angle of 90° was used in the size distribution analysis. All measurements were carried out at 298.0 K. At least six measurements were done for each sample and the average value (standard deviation) was considered.

### 2.5. Encapsulation Efficiency (EE)

The EE was calculated by using the following equation:(1)$$\%\;LEV\;encapsulated=\;\left(1-\frac{m_{LEV}^{supernatant}}{m_{LEV}^{total}}\right)\times 100$$
where $m_{LEV}^{supernatant}$ is the amount of LEV left in the supernatant when the nanoparticles loaded with the drug were removed and $m_{LEV}^{total}$ is the total amount of LEV added in the preparation. $m_{LEV}^{supernatant}$ was determined by UV-vis spectroscopy, measuring the absorbance at 288 nm.

### 2.6. Transmission Electron Microscopy (TEM)

Transmission electron microscopy experiments were performed with Zeiss Libra 120 equipment. The samples were prepared as follows: A drop of an aqueous solution of polymeric nanoparticles loaded with LEV in PBS buffer pH = 7.4 ([NP] = 1 mg·mL^−1^) was deposited on a cooper grid coated with a carbon film and was air dried at room temperature. Following this, it was stained with uranyl acetate and dried before measurement.

### 2.7. In Vitro Drug Release

The drug-loaded nanoparticles were suspended in PBS buffer (pH = 7.4) in a glass vial under continuous magnetic stirring (200 rpm) at 37.4 °C. The concentration of LEV-loaded nanoparticles was in the range 0.55 to 1.1 mg·mL^−1^ in all release experiments. At determined time intervals, a sample was removed, and subsequently centrifuged (13,500 rpm, 30 min) and replaced with an equal quantity of PBS buffer. In this way, the drug concentration in the release medium was diluted to mimic the in-vivo removal into the systemic circulation. The samples were appropriately diluted and the LEV concentration was determined using a UV-visible spectrophotometer at 288 nm. The experimental values are the average of three replicates the precision being within 7%. The glass vial was protected from light to avoid possible photodegradation of LEV. The authors checked that, in the absence of light, the LEV was stable for more than two weeks.

### 2.8. In Vitro Antimicrobial Activity

In vitro antimicrobial activities of the loaded nanoparticles were determined on the basis of the MIC values. The MIC is defined as the lowest concentration of an antimicrobial species required to inhibit or kill a microorganism; that is, the minimal concentration without visible growth. A two-fold serial antimicrobial macro-broth dilution method was used [40] and the experiments were done in duplicate and they were reproducible within a precision of better than 8%. The loaded nanoparticles were suspended in Mueller–Hinton broth (from Difco, prepared following the manufacturer´s instructions) to twice the final desired concentration and then serially two-fold diluted. The final concentrations of the loaded nanoparticles in this test were half of those of the initial dilution series because of the addition of an equal volume of inoculum in broth. The inoculum of each bacterial strain was prepared by adjusting the turbidity of a broth culture incubated for a short time to match the McFarland 0.5 turbidity standard [40] and then further diluting it 1:200 in broth. The final bacterial concentration achieved in each test tube was ca. 5 × 10^4^ to 5 × 10^5^ CFU mL^−1^. The cultures were incubated at 310 K for 16 to 20 h. Mueller–Hinton broth without the loaded nanoparticles but inoculated served as a positive control, and tubes with Mueller–Hinton broth containing the loaded nanoparticles but without inoculum were used as negative control. The growth of the microorganisms was determined visually after incubation. A very faint haziness or a small clump of microorganisms was disregarded, whereas a large cluster of growth or definite turbidity was considered evidence that the loaded nanoparticles failed to completely inhibit growth at that concentration. The lowest concentration of loaded nanoparticles resulting in incomplete inhibition of visible growth was taken as the MIC [40]. In order to calculate this value, the EE of the nanoparticles was considered.

The whole procedure was also carried out for all the bacterial strains tested with nanoparticles obtained following the same method of preparation, but without the presence of the antibiotic. Results showed that the nanoparticles without the loaded LEV had no antimicrobial activity.

## 3. Results

### 3.1. Zeta Potential and Size of PLGA and CS Nanoparticles

The zeta potential, size, and polydispersity of the PLGA and CS nanoparticles prepared by the different methods are summarized in Table 1, Table 2 and Table 3. Representative dynamic light scattering (DLS) traces are shown in Figure 1.

All the PLGA NPs were negatively charged and all the CS NPs were positively charged, independently of the method used. 

In the case of the PLGA NPs prepared by the NANOP (PVA, AC) and NANOP (Pluronic, AC)*,* the negative charged of the loaded NPs was low and the NPs easily formed large aggregates in solution. However, when the preparation was carried out in the presence of an acid media, either in the presence of HCl or a buffer at pH = 4, the charge of the NPs was highly negative and no aggregation in solution occurred. The use of PVA or Pluronic as surfactant did not have a practical influence on the nanocarriers charge. With regard to the PLGA prepared by the DESE method, the charge of the NPs was not affected either by the nature of the organic solvent or by the presence of a buffer at pH = 4. The charge of the DESE NPs was high and aggregation in the media was not observed.

In the case of the CS NPs, the two methods used to render nanocarriers did not show tendency to aggregate in solution because they were stabilized by electrostatic repulsions. Although by using the PG/IC method NPs with a more positive charge were obtained as compared to those prepared by the IG method.

In regard to the size of the PLGA NPs, those prepared by NANOP methods were large (size > 400 nm) and the polydispersity was high (>0.6), except in the case of the NANOP (Pluronic, AC, HCl). On the other hand, the use of the DESE method resulted in smaller NPs with a narrower size distribution (see Table 2). In this case, DCM seems the more suitable organic solvent to obtain NPs of an adequate size for parenteral administration (diameter < 400 nm).

The CS NPs obtained by the IG and PC/IG methods were small, with a suitable size for parenteral administration. The size distribution was clearly narrower in the case of the IG method. 

It is worth pointing out that a polydispersity index greater than 0.7 indicates that the sample has a very broad size distribution and the use of the DLS technique would not be adequate. However, the data obtained by this technique for systems with a high polydispersity index have been collected in Table 1 in order to show the broad size distribution present in these formulations. The existence of particles with a large size made them not suitable for parenteral administration.

### 3.2. Encapsulation Efficiency (EE)

The EE values of the NPs prepared by the different methods are also listed in Table 1, Table 2 and Table 3. The EE found for the PLGA NPs prepared by NANOP methods was similar for all the formulations using Pluronic as surfactant. However, this efficiency decreased when the surfactant utilized was PVA. The EE was not much different for the NPs obtained by using both NANOP and DESE methods. The use of ethyl acetate as organic solvent seemed to improve the EE, although an increase in the pH diminished the EE.

The CS NPs obtained by the IG method showed a noticeably low EE. Data in Table 3 clearly highlight that in order to improve the EE the PC/IG method should be used.

### 3.3. In Vitro Drug Release of PLGA and CS Nanoparticles Loaded with LEV

Figure 2 shows the results of the cumulative release obtained for the PLGA and CS NPs prepared by different methods. One can see in this Figure that for the NPs prepared by the NANOP (Pluronic, AC) and PC/IG methods there was initially a rapid discharge of the drug and most of the LEV was quickly released into the solution. The same result was observed for the NPs prepared with NANOP (PVA, AC), NANOP (Pluronic, AC, HCl), and NANOP (Pluronic, AC, pH = 4) methods. This means that changes in the pH of the medium, as well as variations in the surfactant nature, did not practically influence the way the drug was released from the nanocarriers. The release was not studied for the CS IC NPs because of their low EE. For the PLGA NPs prepared using the DESE methods, the observed drug release was slower. The profile of the release was similar when using AC or DCM as organic solvents. The variation of the pH did not affect the release profile.

### 3.4. Transmission Electron Microscopy and In Vitro Antimicrobial Activity of PLGA DESE(DCM) NPs Loaded with LEV

The morphology, as well as the antimicrobial activity, were investigated only for the PLGA NPs obtained by the DESE (DCM) method. This was so because, as will be explained in the Discussion section, these nanocarriers are the more suitable for loading LEV. Figure 3 shows the TEM image of the PLGA DESE (DCM) loaded NPs. The average size of the NPs, measured from several TEM images corresponding to different samples, was ~230 nm. This value is in agreement with that listed in Table 2 within experimental errors. A size distribution was observed, as was expected from the polydispersity value found by DLS. The morphology of the nanocarriers was spherical for all the samples investigated. 

Table 4 summarizes the MIC against several microorganisms for these NPs. MIC was the lowest concentration of NPs needed to inhibit visible growth after 16 to 20 h of incubation at 310 K. The MIC values were estimated using the broth dilution method.

## 4. Discussion

Table 1, Table 2 and Table 3 lists the information obtained in the characterization of the PLGA and CS NPs, prepared by the different methods. Taking into account that the goal is to prepare NPs suitable for parenteral administration, size, and polydispersity are relevant parameters to consider, since it has been shown that the nanocarrier size influences its effectivity to improve the effectiveness of the antibiotic therapeutic [41]. Moreover, the NPs size should be smaller than 400 nm in order to prevent thromboembolism problems. With this in mind, NANOP (Pluronic, AC, HCl) and DESE (DCM) would be suitable methods for preparing PLGA NPs, and both, IG and PC/IG, NANOP methods, for CS NPs. However, the LEV EE of the CS nanocarriers obtained by the IG method was too low for the NPs to be useful.

The study of the in vitro drug release of the antibiotic from the NPs is important because it gives information about how the nanocarriers retain the drug. Figure 2 shows the results of the cumulative release obtained for the PLGA and CS NPs prepared by different methods. As was mentioned above, for NPs prepared by the NANOP Pluronic, PC/IG, NANOP (AC) and NANOP (HCl) methods there was initially a burst release where most of the LEV was absorbed into the solution in a short period of time. The release was not studied for the CS IC NPs because of their low EE. However, in the case of the PLGA, NPs prepared using DESE methods the release of the antibiotic was slow. The initial drug release from the polymeric NPs occurs by diffusion of the LEV from the polymeric matrix to the solution, followed by a subsequent slower release mediated through both degradation of the polymer matrix and diffusion of the drug [30]. Therefore, the results point out that, in the PLGA NPs prepared using NANOP methods and CS NPs, most of the antibiotic is at, or near to, the NPs surface and can be readily released. For the PLGA NPs prepared using the DESE methods, the release of the antibiotic is slower, this meaning that more of the LEV is located within the polymeric matrix of the nanocarriers. Since a progressive release of the drug is an advantage in the therapeutic effect of the antibiotic, the more suitable nanocarriers for LEV are the NPs obtained using the DESE (DCM) method. With this in mind, the study of the morphology of the NPs using TEM as well as the estimation of the MIC against several microorganisms was carried out only with these nanoparticles.

The size and morphology of the PLGA (DESE, DCM) nanoparticles was studied using TEM. The TEM micrograph in Figure 3 shows that the NPs are spherical. As was indicated in the Results section, the size and size distribution observed by microscopy are in agreement with the values listed in Table 2, obtained by DLS. It is important to point out that no populations of PLGA NPs with sizes larger than 400 nm were observed, which means that the prepared nanocarriers had the size requirements for parenteral administration. 

The antimicrobial activity of the PLGA (DESE, DCM) NPs loaded with LEV against Gram-positive and Gram-negative bacteria was expressed as the minimum inhibitory concentration, and the values are summarized in Table 4. This Table also includes the MIC corresponding to the pure LEV, which are in agreement with literature data [42,43,44,45]. Table 4 shows that the antibacterial activity of the NPs was approximately half of that corresponding to pure LEV for all the microorganisms investigated. However, the use of the loaded LEV PLGA nanoparticles highlighted the advantage of the slow release of the antibiotic, which would decrease the side effects of the drug.

Future research will involve the possibility to carry out in vivo assays with the LEV loaded NPs obtained in this work. In fact, new collaborations with biomedical groups have been established with this goal. Besides, the development of methods to prepare suitable polymeric NPs for parenteral administration of the beta-lactamic antibiotic meropenem seems of relevance. This study will also involve the optimization of the preparation method in order to obtain nanocarriers with the required characteristics. This investigation proposes a challenge because of the low stability of meropenem, which explains the limited research in this field in spite of the wide use of this antibiotic in hospitals. 

## 5. Conclusions

In the present work, the method of preparation of LEV loaded polymeric nanoparticles suitable for parenteral administration was optimized. The goal was to find the best nanoformulation with the highest EE, adequate size, and appropriate antibiotic release. The PLGA nanoparticles obtained with a DESE method and dichloromethane as organic solvent have been shown to have the best qualities. Their charge was high enough to avoid aggregation in solution. Their size and polydispersity were adequate and the EE was sufficiently high. The TEM images agree, with regard to the size and polydispersity, that these PLGA/LEV NPs are suitable for parenteral administration. These nanocarriers have also the advantage of the slow release of the antibiotic, which could help to decrease the side effects of the drug, although many other important characteristics are to be factored in to make this statement. Although, for the microorganisms investigated, the microbial activity of these PLGA/LEV NPs is more or less half of that of the pure drug, the slow release of the antibiotic from the NPs would permit an increase in the time period between administrations as well as decrease the drug side effects.

## Figures and Tables

**Figure 1 pharmaceutics-11-00057-f001:**
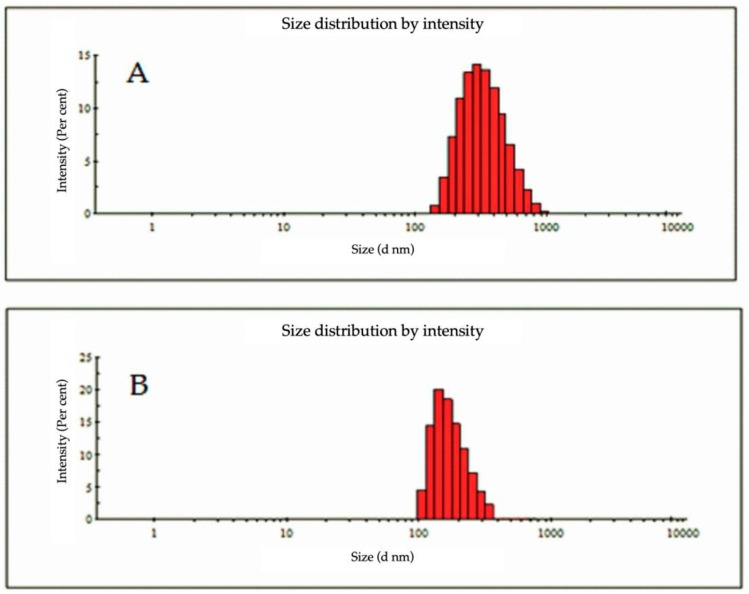
Representative DLS traces. (**A**) double emulsion-solvent evaporation (DESE) (ethyl acetate (EA), pH = 4); (**B**) polyelectrolyte complexation/ionic gelation (PC/IG).

**Figure 2 pharmaceutics-11-00057-f002:**
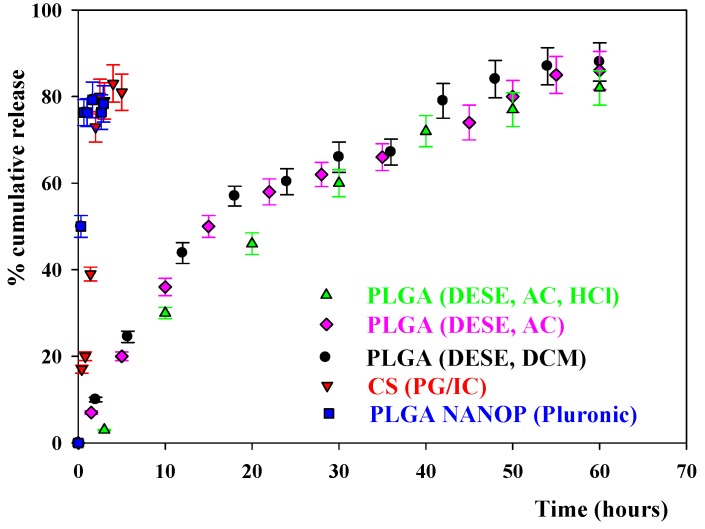
The cumulative release from PLGA and CS nanoparticles prepared by different methods.

**Figure 3 pharmaceutics-11-00057-f003:**
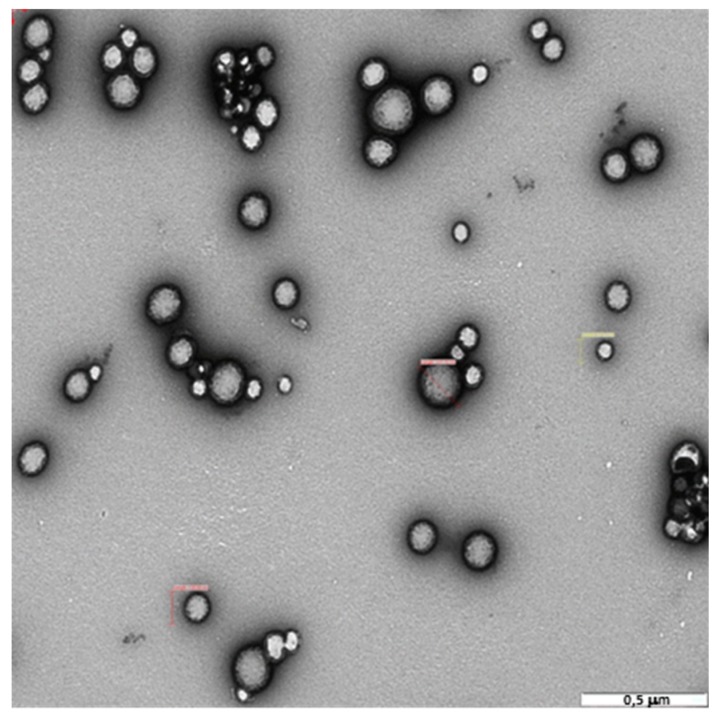
TEM image of PLGA (DESE, DCM) nanoparticles.

**Table 1 pharmaceutics-11-00057-t001:** Characterization of poly(lactic-*co*-glycolic acid) (PLGA) nanoparticles obtained by using nanoprecipitation (NANOP) methods (experimental results are expressed as the mean ± SD, *n* = 6 for DLS measurements and *n* = 3 for encapsulation efficiency (EE)).

Method	Zeta Potential (mV)	Size (nm)	Polydispersity	Encapsulation Efficiency (% LEV)
NANOP (PVA, AC)	−4.3 ± 0.2	430 ± 20	0.61 ± 0.06	16 ± 1
NANOP (Pluronic, AC)	−7.4 ± 0.4	1000 ± 300	0.70 ± 0.09	27 ± 3
NANOP (Pluronic, AC, HCl)	−52 ± 3	222 ± 15	0.42 ± 0.05	27 ± 2
NANOP (Pluronic, AC, pH = 4)	−44 ± 3	950 ± 80	0.91 ± 0.09	21 ± 3

AC: Acetone.

**Table 2 pharmaceutics-11-00057-t002:** Characterization of PLGA nanoparticles obtained by using DESE methods (experimental results are expressed as the mean ± SD, *n* = 6 for DSL measurements and *n* = 3 for EE).

Method	Zeta Potential (mV)	Size (nm)	Polydispersity	Encapsulation Efficiency (% LEV)
DESE (EA)	−33 ± 2	424 ± 23	0.35 ± 0.02	30 ± 3
DESE (EA, pH = 4)	−33 ± 3	424 ± 27	0.35 ± 0.03	13 ± 1
DESE (DCM)	−33 ± 2	221 ± 14	0.35 ± 0.02	22 ± 1

DCM: Dichloromethane.

**Table 3 pharmaceutics-11-00057-t003:** Characterization of chitosan (CS) nanoparticles obtained by using ionic gelation (IG), and PC/IG methods (experimental results are expressed as the mean ± SD, *n* = 6 for DSL measurements and *n* = 3 for EE).

Method	Zeta Potential (mV)	Size (nm)	Polydispersity	Encapsulation Efficiency (% LEV)
IG	18 ± 1	218 ± 11	0.15 ± 0.01	3 ± 1
PC/IG	28 ± 2	108 ± 6	0.45 ± 0.03	25 ± 1

**Table 4 pharmaceutics-11-00057-t004:** Minimum inhibitory concentration (MIC), for the PLGA (DESE, DCM) levofloxacin (LEV) loaded nanoparticles, nanoparticle (NP) LEV, and free antibiotic, LEV, expressed as µg/mL. Experiments were performed in duplicate and the mean MIC value was reported (experimental results are expressed as the mean ± SD, *n* = 3).

MIC (µg/mL)		
Microorganism	NP LEV	LEV
Gram-positive		
*Enterococcus faecalis* CECT 481	4.0 ± 0.3	2.11 ± 0.13
*Mycobacterium phlei* CECT 3009	1.03 ± 0.07	0.25 ± 0.02
*Staphylococcus aureus* ATCC 27697	1.22 ± 0.06	0.50 ± 0.03
Gram-negative		
*Escherichia coli* CECT 101	0.54 ± 0.03	0.25 ± 0.02
*Klebsiella pneumoniae* CECT 143	1.02 ± 0.06	0.25 ± 0.02
*Pseudomonas aeruginosa* ATCC 27853	2.24 ± 0.12	1.31 ± 0.09

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
