# Peer review of "Optimized Preparation of Levofloxacin Loaded Polymeric Nanoparticles"

_pharmaceutics, 2019, doi:10.3390/pharmaceutics11020057_

Round 1
Reviewer 1 Report
The manuscript presents development of PLGA and chitosan nanoparticles for delivery of levofloxacin. The subject is interesting, however the manuscript should be improved before considering for publication:
- It is difficult to assess the real value of the results, because we know nothing about repeatability of the used methods. How many samples of nanoparticles were prepared by each method – the n and SD should be indicated in Table 1 – 3.
- Authors declare that the study has been carried out in order to optimize the preparation of levofloxacin-loaded polymeric nanoparticles by analysis of particular parameters as organic solvent nature, pH of the aqueous phase or temperature. Although all the preparation methods are well described, the results are presented very laconically. Characteristics of the obtained nanoparticles is presented in tables, but there is no discussion that explain how changing of particluar parameters influence size, morphology and encapsulation eficiency.
- Section 2.7 - what was the concentration of nanoparticles in a release medium?
- The release curves for all samples should be presented in Figure 1. Deviation bars should be added.
- Please, check the points in Figure 1 for PLGA NANOP (Pluronic), because this is not the cumulative release.
Author Response
The English
has been revised by a native. - It is difficult to assess the real value of the results, because we know
nothing about repeatability of the used methods. How many samples of
nanoparticles were prepared by each method – the n and SD should be indicated
in Table 1 – 3.y
The number of nanoparticle samples obtained by each method has been indicated in the text (at the end of Section 2.2). The values of n and SD have been included in Tables 1-3.
- Section 2.7 - what was the concentration of nanoparticles in a release medium?
The concentrations of nanoparticles in the different release media have been included in Section 2.7.
- Authors declare that the study has been carried out in order to optimize the preparation of levofloxacin-loaded polymeric nanoparticles by analysis of particular parameters as organic solvent nature, pH of the aqueous phase or temperature. Although all the preparation methods are well described, the results are presented very laconically. Characteristics of the obtained nanoparticles is presented in tables, but there is no discussion that explain how changing of particluar parameters influence size, morphology and encapsulation eficiency.
The text has been modified in order to explain the influence of the different parameters investigated in the characteristics of the NPs obtained.
- The release curves for all samples should be presented in Figure 1. Deviation bars should be added.
Figure 1 has been modified following the reviewers 1 and 2 suggestions. Not all the data corresponding to the PLGA NANOP and CS IG PC/IG have been included because the Figure would be difficult to understand. The deviation bars have been added.
- Please, check the points in Figure 1 for PLGA NANOP (Pluronic), because this is not the cumulative release.
Data have been checked.

Reviewer 2 Report
In this paper Lopez-Lopez et al. evaluate various preparative methods for the manufacturing of polymeric nanoparticles as vehicles for encapsulation and controlled delivery of antibiotic levofloxacine. The quality of the nanoformulations is evaluated in terms of size/dispersity, encapsulation efficiency, drug release, and bacterial growth inhibition assay.
The research done in this work although still exploratory is of some general interest; however the manuscript is presented in a rather unpleasant and sloppy manner. In the present for the manuscript is not suitable for publication.
The introduction (and in general the manuscript as a whole) fails to include a critical assessment of relevant current preparative methods of nanoparticles for the encapsulation of levofloxacine (or parent compounds), above all those based on polyesters and/or chitosan, which are the object of the work. The authors state that “a systematic study in order to improve the nanocarrier characteristics and the yield of encapsulation has not been carried out to date”: Although this might be true by no means does it exclude a myriad of papers dedicated to encapsulation and delivery of levofloxacine. Are the authors the first to report encapsulation and controlled release of this molecule? The Introduction should also include another section discussing the advantage and disadvantage of using different nanoparticle preparation methods above all for the encapsulation of different cargos (which one is more suitable for a given molecule, or which macromolecular parameters are necessary for particle formation). For instance it is not straightforward for a non expert to understand chitosan-based nanoparticles (often used for encapsulation of nucleic acids) have been chosen to encapsulate LEV? Same for PLGA (often used with more hydrophobic drugs). Or what are the advantages/disadvantages of nanoemulsion-solvent evaporation vs nanoprecipitation? These points should be addressed.
There is no comment/discussion on how the preparation method may affect particle characteristics: why is there better size and polydispersity in some particles?, why and how different methods provide better/worse release data?, etc.
Also, the analysis presented is the manuscript is a weak in some parts. For instance, the authors claim to have investigated the effect of pH on the encapsulation of the drug but no part of the manuscript addresses the most important results of this study. Same for solvent: how different solvents affect the particle parameters? Which solvents are better for encapsulation, which ones are better removed, etc. How a more or less acidic pH would affect encapsulation?
Some sections of the manuscript must be rewritten. The authors organise the manuscript into separate “results” and ”discussion” sections, however, little is commented in some parts of the “results” section, apart from a mere reference to the figures and graphs, with no quantitative (or qualitative) analysis to the results obtained. To mention a few, results of TEM analysis make no mention of morphological parameters or dispersity of the particles. Or in the discussion of the different preparative methods for the synthesis of the particles there is no analysis of any kind of the suitability of each method in terms of size, z-potential or EE, where the authors skip through this and just state that encapsulation data are not good for one type of particles. In other instances results alongside discussion of such results are intertwined in the same section. L241-L247 is clearly discussion of the results but it has been included in the “results” section.
The use of DLS for the analysis of some of the formulations seems questionable (too big, too polydisperse). The authors should include representative DLS traces of all of the formulations.
The authors state “that future work will involve the development of methods to prepare suitable nanocarriers for parental administration of the beta-lactamic antibiotic meropenem”. Then, what is the purpose of the present work (focused on LEV)? Do the authors consider the preparative procedure of the particles have been fully optimised? If so, do the authors consider the current particles good enough (or not) to be tested in in vivo models?
All of the above make the manuscript broadly confusing to the reader.
Other points:
Inconsistent use of abbreviations: CS, LEV, DESE.
Typos: glicolic (L72), carraagenan (L141), sisytemic (L180), L414 (bioativity)
Why is there TEM analysis of only one type of particles? How many particles have been analysed for the size measurement by TEM? What is the conclusion of the morphological evaluation of the particles by TEM?
What is the molecular weight of the CS employed? Same for carrageenan (not included in materials section).
L75: seven or six? Six in table 4.
L90: “same polymer mixture” is actually twice diluted of what is stated in L84, isn’t it?
Gives unreliable data or is not possible altogether. Include DLS results.
Ref 31 contains a typo.
L91: that is not the same polymer mixture, it is half concentrated.
L156: “more than one day” – does this cover the entire preparative procedure and the time scale of the drug release data (70 h)?
L172: was TEM recorded of particles in PBS? What molarity? Didn’t this affect the quality of the images (i.e. formation of salt crystals). What was the concentration of the particles and/or LEV in this experiment?
L176: concentration of PBS.
L216: “tendency to aggregate” – this without indication of the medium and time scale gives no information. PC/IG particles have a relatively large potential, which would point to a rather high stability. Why are these particles not stable?
Table 1: why the size numbers are in different format? Also, for the majority of NANOP formulations it is questionable if DLS is the best technique to analyse such polydispersity formulations. It seems to me that the DLS instrument would not be able to do a good cumulat fitting for particles with polydispersity values of 0.7 or higher, especially with sizes in the micron range. If this is the case, the numbers obtained are unreliable. The authors should comment on this. If the fitting is not good, why is this data included in the table?
L225-L230: please reword this paragraph. Include some quantitative data analysis. Just stating higher or more polydispersity is not enough. What is an acceptable size, what is an acceptable polydispersity, are the values obtained in range?
L250-L254: Any comment on the results obtained?
Figure 1: Why are these three separate graphs? The differences between the three formulations would be better seen if the three graphs were merged, or at least represent the top and bottom graphs with the same horizontal scale.
L265-L272: Is potential also important? Is EE also important? Which nanoparticles satisfy the 400 nm cu-off? Which particles present good potential data?
L308-L309: Please indicate in which way TEM images show the suitability of the particles for parenteral administration.
Author Response
The English has been revised by a native.
-The introduction (and in general the manuscript as a whole) fails to include a critical assessment of relevant current preparative methods of nanoparticles for the encapsulation of levofloxacine (or parent compounds), above all those based on polyesters and/or chitosan, which are the object of the work. The authors state that “a systematic study in order to improve the nanocarrier characteristics and the yield of encapsulation has not been carried out to date”: Although this might be true by no means does it exclude a myriad of papers dedicated to encapsulation and delivery of levofloxacine. Are the authors the first to report encapsulation and controlled release of this molecule? The Introduction should also include another section discussing the advantage and disadvantage of using different nanoparticle preparation methods above all for the encapsulation of different cargos (which one is more suitable for a given molecule, or which macromolecular parameters are necessary for particle formation). For instance it is not straightforward for a non expert to understand chitosan-based nanoparticles (often used for encapsulation of nucleic acids) have been chosen to encapsulate LEV? Same for PLGA (often used with more hydrophobic drugs). Or what are the advantages/disadvantages of nanoemulsion-solvent evaporation vs nanoprecipitation? These points should be addressed.
The Introduction section has been modified and new references have been included in order to consider the reviewer comments and suggestions.
-There is no comment/discussion on how the preparation method may affect particle characteristics: why is there better size and polydispersity in some particles?, why and how different methods provide better/worse release data?, etc.
Also, the analysis presented is the manuscript is a weak in some parts. For instance, the authors claim to have investigated the effect of pH on the encapsulation of the drug but no part of the manuscript addresses the most important results of this study. Same for solvent: how different solvents affect the particle parameters? Which solvents are better for encapsulation, which ones are better removed, etc. How a more or less acidic pH would affect encapsulation?
The influence of the different parameters on the particle characteristics has been discussed.
-Some sections of the manuscript must be rewritten. The authors organise the manuscript into separate “results” and ”discussion” sections, however, little is commented in some parts of the “results” section, apart from a mere reference to the figures and graphs, with no quantitative (or qualitative) analysis to the results obtained. To mention a few, results of TEM analysis make no mention of morphological parameters or dispersity of the particles. Or in the discussion of the different preparative methods for the synthesis of the particles there is no analysis of any kind of the suitability of each method in terms of size, z-potential or EE, where the authors skip through this and just state that encapsulation data are not good for one type of particles. In other instances results alongside discussion of such results are intertwined in the same section. L241-L247 is clearly discussion of the results but it has been included in the “results” section.
The reason of writing Results and Discussion as separate sections was in order to adequate the manuscript to the template provided by the journal.
-The authors state “that future work will involve the development of methods to prepare suitable nanocarriers for parental administration of the beta-lactamic antibiotic meropenem”. Then, what is the purpose of the present work (focused on LEV)? Do the authors consider the preparative procedure of the particles have been fully optimised? If so, do the authors consider the current particles good enough (or not) to be tested in in vivo models?
The research carried out in this work gave the authors information about the effect of varying several parameters in the formulations on the characteristics of the prepared loaded levofloxacin nanocarriers. To say that the preparative method is fully optimized seems too optimistic. However, the information provided could be used for the encapsulation of other hydrophilic antibiotics such as meropenem.
The authors have considered the possibility to carry out in vivo experiments with the LEV NPs and other nanocarriers prepared in future works. In fact, a new collaboration with Dr. Ivan Valle from the Institute of Biomedicine of the University of Seville (IBIS) has been established with this goal. This collaboration is at its first stages.
-The use of DLS for the analysis of some of the formulations seems questionable (too big, too polydisperse). The authors should include representative DLS traces of all of the formulations.
Figure 1 showing representative DLS traces has been included in the text.
Other points
Inconsistent use of abbreviations: CS, LEV, DESE?
The meaning of the abbreviations was indicated the first time they were used.
- Typos: glicolic (L72), carraagenan (L141), sisytemic (L180), L414 (bioativity)
The typing mistakes have been corrected.
-Why is there TEM analysis of only one type of particles? How many particles have been analysed for the size measurement by TEM? What is the conclusion of the morphological evaluation of the particles by TEM?
The reason for using TEM only for one type of the nanoparticles obtained was indicated in the section 3.3 as well as in the Discussion section. The comments about the TEM results are included in the text in the Discussion section (L-359-L-361).
Several TEM images were registered from the prepared samples and similar results, in regard to the size and morphology of the PLGA nanoparticles investigated, were observed. Figure 2 is one of the images obtained.
- What is the molecular weight of the CS employed? Same for carrageenan (not included in materials section).
The information has been included in the Materials section.
- L75: seven or six? Six in table 4.
The typing mistake has been corrected.
- Ref 31 contains a typo.
The typo has been corrected.
- L91: that is not the same polymer mixture, it is half concentrated.
It is the same polymer mixture (30 mg of PLGA in 1 mL of DCM). There was a typing mistake (2 mL instead of 1 mL) which has been corrected.
- L156: “more than one day” – does this cover the entire preparative procedure and the time scale of the drug release data (70 h)?
In the description of the methods, one can read that the time scale of the entire preparative procedure is less than 24 h. In the case of the drug release experiments, the glass vial used was, in all cases, protected from light in order to avoid any possible levofloxacin degradation. The authors previously checked that in the absence of light the levofloxacin was stable for more than two weeks. This point has been included in Section 2.3.
- L172: was TEM recorded of particles in PBS? What molarity? Didn’t this affect the quality of the images (i.e. formation of salt crystals). What was the concentration of the particles and/or LEV in this experiment?
The TEM images were recorded using nanoparticles dispersed in PBS 0.01 M pH=7.4. As can be seen in the TEM image no formation of crystals was observed. The concentration of the nanoparticle dispersion has been indicated in Section 2.6.
- L176: concentration of PBS.
The concentration of PBS has been included in the text (Section 2.1).
- L216: “tendency to aggregate” – this without indication of the medium and time scale gives no information. PC/IG particles have a relatively large potential, which would point to a rather high stability. Why are these particles not stable?
The authors have clarified the term “tendency to aggregate” in the text. On the other hand, the authors never say that the PC/IG nanoparticles are unstable.
- Table 1: why the size numbers are in different format? Also, for the majority of NANOP formulations it is questionable if DLS is the best technique to analyse such polydispersity formulations. It seems to me that the DLS instrument would not be able to do a good cumulat fitting for particles with polydispersity values of 0.7 or higher, especially with sizes in the micron range. If this is the case, the numbers obtained are unreliable. The authors should comment on this. If the fitting is not good, why is this data included in the table?
The size numbers have been expressed in the same format. A paragraph in relation to the applicability of DLS for a system with a polydispersity index greater than 0.7 has been added at the end of Section 3.1. An explanation of why the results obtained in such circumstances are shown is also given.
- L225-L230: please reword this paragraph. Include some quantitative data analysis. Just stating higher or more polydispersity is not enough. What is an acceptable size, what is an acceptable polydispersity, are the values obtained in range?
The paragraph has been rewritten and quantitative data have been included. In relation to the size and polydispersity, an acceptable value depends on the use of the nanoparticles. If the particles are going to be used for parenteral administration of a drug the acceptable size is less than 400 nm. If they are used to do investigations on monodisperse formulations the polidispersity should be as small as possible. High polydispersity indexes can be due to aggregation/agglomeration, but also simply by different primary size.
- L250-L254: Any comment on the results obtained?
The text has been modified in order to comment the results obtained.
- Figure 1: Why are these three separate graphs? The differences between the three formulations would be better seen if the three graphs were merged, or at least represent the top and bottom graphs with the same horizontal scale.
Figure 1 has been modified following the reviewers 1 and 2 suggestions. Not all the data corresponding to the PLGA NANOP and CS IG PC/IG have been included because the Figure would be difficult to understand.
- L265-L272: Is potential also important? Is EE also important? Which nanoparticles satisfy the 400 nm cu-off? Which particles present good potential data?
As was indicated in the text, the better nanoparticles obtained from the different formulations used are those prepared by the DESE(DCM) method. Their charge is high enough to avoid aggregation in solution, the size and polydispersity are adequate for parenteral administration and the encapsulation efficiency is sufficiently high. The TEM images also show that these PLGA/LEV NPs are suitable for parenteral administration and they have the advantage of the slow release of the antibiotic, which would decrease the side effects of the drug. Although, for the microorganisms investigated, the microbial activity of these PLGA/LEV NPs is more or less half of that of the pure drug, the slow release of the antibiotic from the NPs would permit to increase the time period between administrations as well as to decrease the drug side effects. The text of the Conclusion section has been modified to consider these points.
- L308-L309: Please indicate in which way TEM images show the suitability of the particles for parenteral administration.
TEM images show that the size and polydispersity of the NPs obtained by the DESE(DCM) method are suitable for parenteral administration since they are small enough (< 400 nm) and with an adequate size distribution.

Round 2
Reviewer 1 Report
The manuscript has been significantly improved and I recommend it for consideration for publication. There is one issue unresolved - the release profile of PLGA NANOP (Pluronic) in Figure 2 is not the cumulative release (some points show decreased release with time).
Author Response
The English has been revised
by a native. Comments and Suggestions for
Authors -The manuscript has been significantly improved and I recommend it for
consideration for publication. There is one issue unresolved - the release
profile of PLGA NANOP (Pluronic) in Figure 2 is not the cumulative release
(some points show decreased release with time). The release
of PGLA NANOP (Pluronic) have been revised and the data in Figure 2 have been
corrected.

Reviewer 2 Report
The authors have answered the requests from the reviewers in an overall satisfactory fashion. The manuscript has been improved considerably, so as to be ready for publication after minor corrections.
I feel like the author missed the points of some of my comments. Apologies if I did not explain myself correctly, hopefully the below will be more clear.
non consistent use of abbreviations: CS, LEV, DESE?
The meaning of the abbreviations was indicated the first time they were used.
This is correct, however, such abbreviations have not been used consistently afterwards: chitosan, levofloxacin and double emulsion-solvent evaporation method have been used afterwards instead of the relevant abbreviation.
-The authors state “that future work will involve the development of methods to prepare suitable nanocarriers for parental administration of the beta-lactamic antibiotic meropenem”. Then, what is the purpose of the present work (focused on LEV)? Do the authors consider the preparative procedure of the particles have been fully optimised? If so, do the authors consider the current particles good enough (or not) to be tested in in vivo models?
The research carried out in this work gave the authors information about the effect of varying several parameters in the formulations on the characteristics of the prepared loaded levofloxacin nanocarriers. To say that the preparative method is fully optimized seems too optimistic. However, the information provided could be used for the encapsulation of other hydrophilic antibiotics such as meropenem.
The authors have considered the possibility to carry out in vivo experiments with the LEV NPs and other nanocarriers prepared in future works. In fact, a new collaboration with Dr. Ivan Valle from the Institute of Biomedicine of the University of Seville (IBIS) has been established with this goal. This collaboration is at its first stages.
No doubt the encapsulation of meropenem is of relevance to the authors, however, I would consider equally or more relevant for this paper if the authors disclose that they are dedicating efforts to continue the current work (as they state with in vivo experiments), that is, with the particles developed in this paper. It is at the authors’ discretion whether they want to disclose this or not.
-The TEM images also show that these PLGA/LEV NPs are suitable for parenteral administration and they have the advantage of the slow release of the antibiotic, which would decrease the side effects of the drug.
This sentence is misleading: the TEM images show nanoparticles of a certain dispersity and size. Thats it. No conclusion about release can be drawn from TEM images, which could be interpreted as a conclusion the way the sentence is drafted. Also, the adequacy of parenteral administration or the appearance of side effects cannot be judged/predicted alone on particle size/dispersity and slow drug release, many other important characteristics are to be factored in to make this statement. This sentence must be redraft accordingly.
-Some sections of the manuscript must be rewritten. The authors organise the manuscript into separate “results” and ”discussion” sections, however, little is commented in some parts of the “results” section, apart from a mere reference to the figures and graphs, with no quantitative (or qualitative) analysis to the results obtained. To mention a few, results of TEM analysis make no mention of morphological parameters or dispersity of the particles. Or in the discussion of the different preparative methods for the synthesis of the particles there is no analysis of any kind of the suitability of each method in terms of size, z-potential or EE, where the authors skip through this and just state that encapsulation data are not good for one type of particles. In other instances results alongside discussion of such results are intertwined in the same section. L241-L247 is clearly discussion of the results but it has been included in the “results” section.
The reason of writing Results and Discussion as separate sections was in order to adequate the manuscript to the template provided by the journal.
My point was not related to why the manuscript was divided into two separate “results” and “discussion” section. My point is that the “result” section in some case does not actually disclose the relevant results, or it does contain parts that would be better included in the “discussion” section.
For instance, the authors do not disclose the average size by TEM. How TEM be in aggrement with DLS if the average size by TEM is not reported? Typically, hydrophilic particles exhibit a smaller size by TEM than DLS. Is this the case? How about other parameters that can be studied from the TEM images: shape of the particles, dispersity, etc.? Can the authors comment on this from the TEM images? All the authors have commented on is that particles by TEM are below 400 nm, but much more can be said.
Also, the size is scale in the TEM image is barely visible. Can this be corrected?
Taking into account that the goal is to prepare NPs suitable for parenteral administration, size and polydispersity are relevant parameters to consider, since it has been shown that the nanocarrier size influences on its effectivity to improve the antibiotic therapeutic effectivity [43]. Besides, the NPs size should be smaller than 400 nm in order to prevent thromboembolism problems. With this in mind, NANOP(Pluronic, AC, HCl) and DESE(DCM) would be suitable methods for preparing PLGA NPs, and both, IG and PC/IG nanoprecipitation methods, for CS NPs. However, the LEV encapsulation efficiency of the CS nanocarriers obtained by the IG method is too low for the NPs to be useful.
Can the authors state here what is their take on an acceptable polydispersity for a parenteral nanoformulation? Can the authors include here the actual values of size, dispersity and EE of the formulations that are in their view suitable for parental formulation, rather than just making a mention to the relevant table?
I think Figure 1 may benefit from using of different symbols in addition to the already used different colours to differentiate the different formulation. The authors could see if this further improves the visibility of the data.
L151: the yield of the method? does this refer to EE?
L175: Carrageenan
L542 reference contains a typo. Check all references for additional typos.
.
Author Response
The English has been revised by a native.
Comments and Suggestions for Authors
-The authors have answered the requests from the reviewers in an overall satisfactory fashion. The manuscript has been improved considerably, so as to be ready for publication after minor corrections.
I feel like the author missed the points of some of my comments. Apologies if I did not explain myself correctly, hopefully the below will be more clear.
non consistent use of abbreviations: CS, LEV, DESE?
The meaning of the abbreviations was indicated the first time they were used.
This is correct, however, such abbreviations have not been used consistently afterwards: chitosan, levofloxacin and double emulsion-solvent evaporation method have been used afterwards instead of the relevant abbreviation.
Following the reviewer suggestion the abbreviations were used throghout the text of the manuscript.
-The authors state “that future work will involve the development of methods to prepare suitable nanocarriers for parental administration of the beta-lactamic antibiotic meropenem”. Then, what is the purpose of the present work (focused on LEV)? Do the authors consider the preparative procedure of the particles have been fully optimised? If so, do the authors consider the current particles good enough (or not) to be tested in in vivo models?
The research carried out in this work gave the authors information about the effect of varying several parameters in the formulations on the characteristics of the prepared loaded levofloxacin nanocarriers. To say that the preparative method is fully optimized seems too optimistic. However, the information provided could be used for the encapsulation of other hydrophilic antibiotics such as meropenem.
The authors have considered the possibility to carry out in vivo experiments with the LEV NPs and other nanocarriers prepared in future works. In fact, a new collaboration with Dr. Ivan Valle from the Institute of Biomedicine of the University of Seville (IBIS) has been established with this goal. This collaboration is at its first stages.
-No doubt the encapsulation of meropenem is of relevance to the authors, however, I would consider equally or more relevant for this paper if the authors disclose that they are dedicating efforts to continue the current work (as they state with in vivo experiments), that is, with the particles developed in this paper. It is at the authors’ discretion whether they want to disclose this or not.
The text has been modified to consider the reviewer suggestion.
-The TEM images also show that these PLGA/LEV NPs are suitable for parenteral administration and they have the advantage of the slow release of the antibiotic, which would decrease the side effects of the drug.
This sentence is misleading: the TEM images show nanoparticles of a certain dispersity and size. Thats it. No conclusion about release can be drawn from TEM images, which could be interpreted as a conclusion the way the sentence is drafted. Also, the adequacy of parenteral administration or the appearance of side effects cannot be judged/predicted alone on particle size/dispersity and slow drug release, many other important characteristics are to be factored in to make this statement. This sentence must be redraft accordingly.
The text has been modified.
-Some sections of the manuscript must be rewritten. The authors organise the manuscript into separate “results” and ”discussion” sections, however, little is commented in some parts of the “results” section, apart from a mere reference to the figures and graphs, with no quantitative (or qualitative) analysis to the results obtained. To mention a few, results of TEM analysis make no mention of morphological parameters or dispersity of the particles. Or in the discussion of the different preparative methods for the synthesis of the particles there is no analysis of any kind of the suitability of each method in terms of size, z-potential or EE, where the authors skip through this and just state that encapsulation data are not good for one type of particles. In other instances results alongside discussion of such results are intertwined in the same section. L241-L247 is clearly discussion of the results but it has been included in the “results” section.
The reason of writing Results and Discussion as separate sections was in order to adequate the manuscript to the template provided by the journal.
My point was not related to why the manuscript was divided into two separate “results” and “discussion” section. My point is that the “result” section in some case does not actually disclose the relevant results, or it does contain parts that would be better included in the “discussion” section.
The text has been modified in order to consider the reviewer comments. For this reason, the order of the references 42 and 43 has been exchanged.
-For instance, the authors do not disclose the average size by TEM. How TEM be in aggrement with DLS if the average size by TEM is not reported? Typically, hydrophilic particles exhibit a smaller size by TEM than DLS. Is this the case? How about other parameters that can be studied from the TEM images: shape of the particles, dispersity, etc.? Can the authors comment on this from the TEM images? All the authors have commented on is that particles by TEM are below 400 nm, but much more can be said.
An average size value was estimated from several images obtained for different samples. The text has been changed following the reviewer comments.
-Also, the size is scale in the TEM image is barely visible. Can this be corrected?
The size scale in Figure 3 has been modified.
-Taking into account that the goal is to prepare NPs suitable for parenteral administration, size and polydispersity are relevant parameters to consider, since it has been shown that the nanocarrier size influences on its effectivity to improve the antibiotic therapeutic effectivity [43]. Besides, the NPs size should be smaller than 400 nm in order to prevent thromboembolism problems. With this in mind, NANOP(Pluronic, AC, HCl) and DESE(DCM) would be suitable methods for preparing PLGA NPs, and both, IG and PC/IG nanoprecipitation methods, for CS NPs. However, the LEV encapsulation efficiency of the CS nanocarriers obtained by the IG method is too low for the NPs to be useful.
Can the authors state here what is their take on an acceptable polydispersity for a parenteral nanoformulation? Can the authors include here the actual values of size, dispersity and EE of the formulations that are in their view suitable for parental formulation, rather than just making a mention to the relevant table?
About the suitable size for parenteral administration, as is indicated in the text, the nanocarriers should be smaller than 400 nm in order to avoid thromboembolism problems. Therefore, the size distribution needs to be narrow enough not to expect a certain population of nanocarriers larger than this size. In regard to the EE, that would depend on the dosification; that is, on the drug and the illness in cuestion. Besides, in this regard, in vivo studies would be necessary in order to investigate the effectiveness of using the nanocarriers to treat the illness. The text has been modified in order to comment about the polydispersity requirements.
-I think Figure 1 may benefit from using of different symbols in addition to the already used different colours to differentiate the different formulation. The authors could see if this further improves the visibility of the data.
The symbols in the Figure have been changed following the reviewer suggestion.
-L151: the yield of the method? does this refer to EE?
-L175: Carrageenan
-L542 reference contains a typo. Check all references for additional typos.
The three comments have been considered in the text.

Round 3
Reviewer 2 Report
The manuscript is ready for publication